# Molecular Characterization of Three Tandemly Located Flagellin Genes of *Stenotrophomonas maltophilia*

**DOI:** 10.3390/ijms23073863

**Published:** 2022-03-31

**Authors:** Cheng-Mu Wu, Hsin-Hui Huang, Li-Hua Li, Yi-Tsung Lin, Tsuey-Ching Yang

**Affiliations:** 1Department of Biotechnology, Laboratory Science in Medicine, National Yang Ming Chiao Tung University, Taipei 112, Taiwan; wuchengshiou@gmail.com (C.-M.W.); toe3273917@outlook.com (H.-H.H.); 2Department of Pathology, Laboratory Medicine, Taipei Veterans General Hospital, Taipei 112, Taiwan; lilh@vgtpe.gov.tw; 3Program in Medical Biotechnology, Taipei Medical University, Taipei 110, Taiwan; 4Division of Infectious Diseases, Department of Medicine, Taipei Veterans General Hospital, Taipei 112, Taiwan; ytlin8@vghtpe.gov.tw; 5Department of Medicine, National Yang Ming Chiao Tung University, Taipei 112, Taiwan

**Keywords:** flagellin, swimming, *Stenotrophomonas maltophilia*

## Abstract

*Stenotrophomonas maltophilia* is a motile, opportunistic pathogen. The flagellum, which is involved in swimming, swarming, adhesion, and biofilm formation, is considered a virulence factor for motile pathogens. Three flagellin genes, *fliC1*, *fliC2*, and *fliC3*, were identified from the sequenced *S. maltophilia* genome. *FliC1*, *fliC2*, and *fliC3* formed an operon, and their encoding proteins shared 67–82% identity. Members of the *fliC1C2C3* operon were deleted individually or in combination to generate single mutants, double mutants, and a triple mutant. The contributions of the three flagellins to swimming, swarming, flagellum morphology, adhesion, and biofilm formation were assessed. The single mutants generally had a compromise in swimming and no significant defects in swarming, adhesion on biotic surfaces, and biofilm formation on abiotic surfaces. The double mutants displayed obvious defects in swimming and adhesion on abiotic and biotic surfaces. The flagellin-null mutant lost swimming ability and was compromised in adhesion and biofilm formation. All tested mutants demonstrated substantial but different flagellar morphologies, supporting that flagellin composition affects filament morphology. Bacterial swimming motility was significantly compromised under an oxidative stress condition, irrespective of flagellin composition. Collectively, the utilization of these three flagellins for filament assembly equips *S. maltophilia* with flagella adapted to provide better ability in swimming, adhesion, and biofilm formation for its pathogenesis.

## 1. Introduction

Bacteria are capable of sensing environmental changes through highly adapted signaling machinery and can direct their chemotactic movement to more favorable environments by changing their motility [1]. Flagella are one of the most complex and effective structures responsible for the motility of bacteria [2]. In addition to motility, the role of the flagellum in bacterial pathogenicity has been proposed. The flagellum is regarded as a virulence factor that participates in the adherence and invasion of host cells, as well as in biofilm formation [3,4,5,6]. Furthermore, the flagellum has been viewed as a potent immune activator, triggering both innate and adaptive immune responses by host cells during microbial infections [7]. Given these features, flagellin has been proposed as a vaccine candidate to prevent pathogen infection [8] or as a promising candidate for the rapid detection of pathogens in clinical specimens [9].

Because of the antigenicity of flagellin, the expression of flagellin during the course of an infection could be deleterious to pathogens due to flagellin-mediated host recognition. It has been reported that pathogens possess a regulatory system capable of sensing the host and modulating the expression of flagellin. During the infective process, pathogens downregulate the expression of flagellin to escape recognition by host cells and are subsequently upregulated after host mortality [10]. Some pathogens that harbor multiple flagellin genes can utilize flagellin phase switching to escape recognition by the host immune system.

The bacterial flagellum is composed of three main components: the basal body, which works as a rotary motor; the filament, which functions as a screw propeller; and the hook, which acts as a universal joint connecting the filament to the motor [11]. Biosynthesis of the flagellum is a highly ordered process dependent upon sequential secretion and assembly of the rod, hook, hook-associated, cap, and flagellin proteins. Flagellin, the primary structural protein of filaments, is synthesized in the cytoplasm as soluble monomers, secreted by a dedicated type III secretion system within the flagellar basal body, and polymerized to form the filaments of bacterial flagella. For the expression of flagellum-associated genes, a hierarchical transcriptional regulatory model is commonly accepted, despite some differences in different bacteria. A three-level hierarchical regulation model was proposed for *Escherichia coli* [12]; however, a four-class regulation model was applied to *Vibrio cholera* [13], *Pseudomonas aeruginosa* [14], and *Stenotrophomonas maltophilia* [15]. A common feature observed in three-level and four-class models is that flagellin gene expression is coordinately regulated by FliA and FlgM. FliA (σ^28^) is the key sigma factor that regulates the expression of the flagellin gene, whereas FlgM is an anti-sigma factor that binds with FliA to sequester the association between RNA polymerase core enzyme and FliA [16]. In general, the expression of *fliA* and *flgM* precedes the basal body- and hook-associated genes. Before hook basal body (HBB) assembly, the preceding expressed FliA is bound by FlgM, sequestering FliA from triggering the expression of the flagellin gene(s). Upon completion of the hook basal body (HBB), FlgM is secreted by HBB, and the released FliA activates the transcription of the flagellin gene(s).

The flagellation pattern of a bacterium can be defined by the spatial arrangement and number of flagella present on the cell surface. The arrangement and number of flagellin gene(s) vary in the genomes of motile bacteria. The genomes of *P. aeruginosa* and *Bacillus subtilis* are equipped with a single flagellin gene [8,17]. Specifically, *P. aeruginosa* strains can express one of two structurally distinct flagellin proteins encoded by the *fliC* gene, which are designated as type-a and type-b [8]. In contrast, *Salmonella typhimurium*, with two distinct flagellin genes, undergoes phase variation and utilizes one of the two flagellins at a time to build the filament [18]. In contrast, certain flagellated bacteria possess several flagellin genes, such as the five or six flagellin genes in *Vibrio* species [19]. Bacteria with multiple flagellin genes generally assemble the flagellar filament using all, or at least most, of the flagellins encoded by their genome. The observation that different flagellins exhibit diverse spatial arrangements in filaments has been reported in some bacteria, such as *Caulobacter crescentus*, *Helicobacter pylori*, and *Shewanella putrefaciens* [20,21,22]. However, in certain bacteria, such as *C. crescentus*, multiple flagellins are thought to be redundant, and FljJ flagellin can function as a regulator [23,24]. A more interesting finding in *V.*
*vulnificus* is that two flagellin-homologous proteins (FHPs), FlaE and FlaF, are absent from the flagellar structure but are found in culture supernatants. These FHPs are not involved in filament formation and cellular motility but contribute to biofilm formation [25].

*S. maltophilia*, a Gram-negative, non-fermentative bacillus, is abundant, found in ubiquitous environments with a broad geographical distribution [26]. Adhesion of *S. maltophilia* to abiotic surfaces, such as medical implants and catheters, represents a major risk for hospitalized patients. The treatment of *S. maltophilia* infection is a considerable challenge because this microorganism is intrinsically resistant to multiple antimicrobial agents, including most β-lactams, aminoglycosides, and tetracycline [27]. In addition to antimicrobial agents, the management of *S. maltophilia* infection by blocking bacterial adherence and invasion is an alternative strategy. Surface molecules or structures, such as flagella or fimbrial adhesins, are considered first as targets [28]. *S. maltophilia* possesses polar flagella that confer motility and chemotaxis, facilitate adherence to cells and inanimate surfaces, and contribute to colonization and invasion during the early phases of infection [29,30,31]. The regulation of flagellin gene expression involves c-di-GMP and FleQ [32]. A genome-wide survey revealed three flagellin genes in *S. maltophilia* KJ (GenBank accession number JAIQXD000000): *Smlt2304*, *Smlt2305*, and *Smlt2306*. No information related to the three flagellin genes has been reported to date. In this study, we created nonpolar deletion mutants of the three flagellin genes, either alone or in combination, to genetically investigate their functions.

## 2. Results

### 2.1. Bioinformatics Analysis of Flagellin Genes

A BLASTp search of the *S. maltophilia* KJ genome (GenBank accession number JAIQXD000000) using the FliC protein (PA1092) of *P. aeruginosa* as a query revealed three candidates: Smlt2304, Smlt2305, and Smlt2306. Hereafter, we refer to *Smlt2306*, *Smlt2305*, and *Smlt2304* as *fliC1*, *fliC2*, and *fliC3*, respectively. The genes upstream and downstream of the *fliC1C2C3* cluster were *flgL* (encoding flagellar hook–filament junction protein) and *fliD* (encoding flagellar hook-associated protein). The proteins encoded by *fliC1*, *fliC2*, and *fliC3* were 406, 400, and 406 aa, respectively, and exhibited high similarity over their entire length. The protein identity and similarity of FliC1 and FliC2 were 67% and 76%, of FliC1 and FliC3 were 67% and 76%, and of FliC2 and FliC3 were 82% and 86%, respectively. The three flagellin proteins shared nearly identical N- and C-terminal sequences, whereas the central region was variable in size and primary structure (Figure 1).

The FliC protein of *Salmonella* Typhimurium has previously been crystalized [33,34], allowing the structural positioning of amino acid residues and the identification of five domains. The linear arrangement of the domains is: amino domain (ND0, ND1, DN2), D3, carboxyl domain (CD2, CD1, and CD0) (Figure 1). Compared to those of *Salmonella* Typhimurium FliC, the D3 and CD2 domains of *S. maltophilia* FliCs were shorter.

The short intergenic regions in the *fliC1C2C3* cluster strongly suggest the possibility of a *fliC1-fliC2-fliC3* operon. The presence of a *fliC1C2C3* operon was verified using RT-PCR (Figure 2A). We were curious whether the *fliC1C2C3* cluster was well conserved in all *S. maltophilia* isolates. Based on a whole-genome phylogeny analysis, Mercier-Darty et al. have classified the *S. maltophilia* complex into 21 genogroups [35]. We randomly selected some strains as the representatives for each genogroup and analyzed the flagellin gene numbers in each genome. Among the 35 genomes, 24 harbored the *fliC1C2C3* cluster, 8 strains possessed two flagellin genes, and 2 strains were equipped with one flagellin gene (Appendix A). It seemed that there was no correlation between the flagellin gene numbers and the genogroups in *S. maltophilia*.

Although *fliC1*, *fliC2*, and *fliC3* form an operon, it cannot immediately be ruled out that *fliC2* and *fliC3* may have their own promoter activities. Three DNA segments upstream from the *fliC1*, *fliC2*, and *fliC3* genes (labeled as gray bars in Figure 2A) were cloned into promoter-proving vector pRKXylE [36] to generate plasmids pFliC1_xylE_, pFliC2_xylE_, and pFliC3_xylE_. Catechol 2,3-dioxygenase (C23O) is encoded by *xylE*; thus, C23O activity determination can represent promoter activity of the cloned DNA segment. Plasmids pFliC1_xylE_, pFliC2_xylE_, and pFliC3_xylE_ were individually introduced into wild-type KJ, and the expressed C23O activities were determined. Of the three strains tested, only KJ(pFliC1_xylE_) displayed significant C23O activity (Figure 2B), indicating that the *fliC1C2C3* operon is driven by the promoter upstream of *fliC1*, at least under laboratory conditions.

### 2.2. Impact of Filaments with Different Flagellin Combinations on Swimming, Swarming, Flagellum Morphology, Adhesion, and Biofilm Formation

To characterize the three flagellin genes, *fliC1*, *fliC2*, and *fliC3*, we created a complete set of nonpolar deletion mutants for all combinations, generating KJΔFliC1, KJΔFliC2, KJΔFliC3, KJΔFliC2C3, KJΔFliC1C3, KJΔFliC1C2, and KJΔFliC1C2C3. A lack of polar effects in these constructed mutants was verified by qRT-PCR (Appendix A). The KJ cells completely lost their swimming motility when the three flagellin genes were deleted (Figure 3A). Of the three single-deletion mutants, KJΔFliC3 was most severely compromised in terms of swimming motility (Figure 3A). The swimming motility of the three double-deletion mutants varied in the following order: KJΔFliC1C2 > KJΔFliC1C3 > KJΔFliC2C3. It is worth mentioning that the KJΔFliC2C3 mutant construct almost abolished swimming ability (Figure 3A). Nevertheless, to our surprise, all the tested mutants displayed comparable swarming motility with wild-type KJ (Figure 3B).

Flagella morphology was observed under a transmission electron microscope (TEM). Wild-type flagella were present in KJΔFliC1, but the flagella number of KJΔFliC1 was moderately lower than that of wild-type KJ. KJΔFliC3 possessed significantly shorter flagella (Figure 3C). Flagella morphologies were quite different for the three double mutants. KJΔFliC2C3 was flagellated, similar to wild-type KJ. The flagella of KJΔFliC1C3 were quite short, and the flagella number of KJΔFliC1C3 was obviously decreased (Figure 3C).

In addition to their roles in swimming, flagella are important for adhesion and biofilm formation in several bacteria [4,5,6,28]. Thus, we tested the adhesion ability and biofilm formation of KJ and its derived *fliC*-associated mutants on abiotic and biotic surfaces, respectively. For the abiotic surfaces, the adhesion abilities of the *fliC3* single mutant, three double mutants, and triple mutant were lower than that of wild-type KJ (Figure 3D). However, of all the *fliC*-associated mutants tested, only KJΔFliC1C2C3 produced less-developed biofilms than wild-type KJ on abiotic surfaces (Figure 3E). As for the biotic surfaces, the adhesion abilities of the double mutants and the triple mutant were compromised (Figure 3F), and the biofilm formations of the *fliC3*-associaed mutants (KJΔFliC3, KJΔFliC2C3, KJΔFliC1C3, and KJΔFliC1C2C3) were significantly decreased (Figure 3G).

### 2.3. Impact of Filaments with Different Flagellin Combinations on Swimming Motility under Different Environmental Conditions

The most interesting question we considered was the significance of the presence of three different flagellin genes in a pathogen. It has been pointed out that bacteria harboring multiple flagellin genes can utilize different flagellin combinations to form a filament, which is highly beneficial for their spreading through complex environments [22]; thus, we considered the possibility that diverse flagellin combinations may have different effects on swimming motility in response to various environments. Next, we surveyed the contribution of different flagellin combinations to swimming motility under different environmental conditions. Since no significant swimming motility was observed in mutants KJΔFliC2C3 and KJΔFliC1C2C3 (Figure 3A), they were excluded from the following evaluation. The stressors added included menadione (MD) (to mimic oxidative stress), 2,2′-dipyridyl (DIP) (to mimic iron starvation), and Ficoll (to mimic viscous environments). To determine the optimal concentration of stressors added, a dose–response curve of wild-type KJ swimming motility in different stressor concentrations was firstly assessed. We noticed that the bacterial growth was affected when the concentrations of MD, DIP, and Ficoll were greater than 20 μg/mL, 30 μg/mL, and 20%, respectively. Thus, the stressor concentrations were set at 12 μg/mL for MD, 20 μg/mL for DIP, and 10% for Ficoll to rule out growth-difference-mediated biases. Of the three stressors tested, MD had the most obvious negative impact on swimming motility of the wild-type KJ; DIP and Ficoll did not significantly affect swimming motility (Appendix A). Except for KJΔFliC1C3, the swimming-compromise levels of the tested mutants were comparable to that of the wild-type KJ in the three stressed conditions (Figure 4). KJΔFliC1C3 displayed marginal swimming motility in MD- and DIP-containing swimming agars (Figure 4A,B).

### 2.4. Phylogenetic Analysis of Flagellin

The number of flagellin genes vary among motile bacteria. To investigate the phylogenetic relatedness of multiple flagellin genes in a species, the phylogenetic relationships among the flagellin proteins of different species were examined. The flagellin proteins from the same species were phylogenetically clustered, and the number of flagellin genes in a species was not positively correlated with phylogenetic patterns (Figure 5). These results suggested that multiple flagellin genes in a species appear to result from gene duplication events after speciation, rather than before speciation. Salmonella isolates are classified into thousands of serotypes according to different somatic (O) and flagellar (H) antigenic combinations. The flagellin proteins obtained from the *Salmonella* isolates of different H serotypes hardly affected the pattern of the phylogenetic tree; thus, *Salmonella* Typhimurium 33,676 was selected as a representative (Figure 5).

## 3. Discussion

Homologous genes are widely present in bacterial genomes. The proteins encoded by these homologous genes may display functional redundancy or different biological functions, and some may even have no biological function. The results of this study demonstrated that the three flagellin proteins of *S. maltophilia* do not seem redundant for their roles in swimming motility, flagellum morphology, adhesion, and biofilm formation. By integrating the results of Figure 1 and Figure 3, we summarized the following: (1) Substantial but different flagellar morphologies were observed for KJΔFliC2C3, KJΔFliC1C3, and KJΔFliC1C2 (Figure 3C), indicating that FliC1, FliC2, or FliC3 alone could assemble to form flagella with different morphologies. Overall, the bacterial swimming ability seems to be generally proportional to the number of flagellin proteins assembled in a filament; that is, three-flagellin filament > two-flagellin filament > one-flagellin filament (Figure 3A). (2) Of the three single-deletion mutants (KJΔFliC1, KJΔFliC2, and KJΔFliC3), KJΔFliC3 had the most compromised swimming ability. On the other hand, among the three double-deletion mutants (KJΔFliC2C3, KJΔFliC1C3, and KJΔFliC1C2), KJΔFliC1C2 maintained the best swimming ability (Figure 3A). These observations supported that FliC3 plays a critical role in swimming motility. (3) Based on the observation of KJΔFliC2C3 immobility, it seems that FliC1 contributes little to swimming. However, we noticed that the swimming zone of KJΔFliC3 (FliC1/FliC2-assembled flagella) was larger than that of KJΔFliC1C3 (FliC2-assembled flagella). A similar tendency existed for KJΔFliC2 (FliC1/FliC3-assembled flagella) and KJΔFliC1C2 (FliC3-assembled flagella) (Figure 3A). These observations supported that FliC1 plays an auxiliary role in enhancing the FliC2-filament (or FliC3-filament)-mediated swimming ability when it joins with FliC2 (or FliC3) to assemble filaments. (4) KJΔFliC2C3 displayed wild-type-like flagella in morphology (Figure 3C), but no significant swimming motility was detected (Figure 3A). In contrast, KJΔFliC3 presented short and abnormal flagella, but maintained better swimming motility than KJΔFliC2C3 (Figure 3A,C). Thus, flagellar morphology does not always correlate well with swimming ability. (5) The contribution of flagella to adhesion and biofilm formation, either on abiotic or on biotic surfaces, in *S. maltophilia* clinical isolates have been verified by Di Bonaventura’s group [30]. In this study, the individual contribution of FliC1, FliC2, and FliC3 to adhesion and biofilm formation has been further elucidated. In general, two-flagellin filaments kept comparable adhesion ability with the wild-type filament, except KJΔFliC3 with its compromised adhesion ability on abiotic surfaces (Figure 3D,F). The results of the biofilm formation revealed that the presence of at least one flagellin protein is sufficient to promote biofilm formation on abiotic surfaces (Figure 3E). As for the biofilm formation on biotic surfaces, the *fliC3*-associaed mutants (KJΔFliC3, KJΔFliC2C3, KJΔFliC1C3, and KJΔFliC1C2C3) were significantly decreased (Figure 3G). Furthermore, we also noticed that the biotic-surface biofilms of the double mutants (KJΔFliC2C3 and KJΔFliC1C3) and the triple mutant (KJΔFliC1C2C3) were comparable to that of KJΔFliC3. These observations support that FliC3 seems to play a critical role in biofilm formation on biotic surfaces (Figure 3G).

Swarming motility is the movement of bacteria over a moist, nutrient-rich solid surface [37]. The necessity of flagella for swarming has been reported in several Gram-negative bacteria, such as *Proteus mirabilis*, *Serratia marcescens*, *E. coli*, and *P. aeruginosa* [38]. In general, the flagella-null mutants of Gram-negative bacteria completely or partially lose their swarming motility [39,40,41]. However, several studies have also pointed out that the determinants for bacterial swarming are not limited to those affecting flagellar functions [39,40,41,42]. For example, the chemotaxis system plays a critical role in *E. coli* swarming [42], and the swarming of *P. aeruginosa* is dependent on type IV pili [40]. In this study, we show that a flagella-null mutant exhibits a comparable swarming motility with the wild-type in *S. maltophilia* (Figure 3B). Two possibilities are proposed for this observation: (i) In contrast to previous findings, flagella hardly contribute to swarming in *S. maltophilia*; and (ii) other unidentified systems, such as chemotaxis or type IV pili systems, may dominantly contribute to swarming and shield the contribution of flagella in *S. maltophilia*.

It has been reported that bacteria harbor flagella composed of different flagellins that benefit their motility in a wide range of environmental conditions [22]. Except for KJΔFliC1C3, the swimming-compromise levels in the wild-type KJ and the *fliC*-associated mutants were comparable in the stressed conditions tested (Figure 4). Of the three stressors tested, the swimming motility of *S. maltophilia* was affected most by MD.

Collectively, *S. maltophilia* can utilize three different flagellins to assemble filaments, and the three-flagellin flagellum exhibits a better swimming motility than two- or one-flagellin flagella. A significant feature of *S. maltophilia* is its ability to adhere to host tracheal mucus and cause respiratory tract infections [29]. Antibiotic choice for the treatment of *S. maltophilia* infection is limited due to its intrinsic resistance to several antibiotics [32]. Flagellin appears to be a promising target for the development of an inhibitor to limit flagellum-mediated swimming, adhesion, and biofilm formation. Based on the findings in this study, the simultaneous inhibition of FliC1, FliC2, and FliC3 is the preferred consideration. An inhibitor that targets the conserved N- or C-terminal regions of FliC1, FliC2 and FliC3 or a mixture of inhibitors that target the variable regions of FliC1, FliC2 and FliC3, respectively, can be the choices.

## 4. Materials and Methods

### 4.1. Bacterial Strains, Plasmids, and Primers

The bacterial strains, plasmids, and primers used in this study are listed in Appendix A.

### 4.2. Reverse Transcription PCR

DNA-free RNA from logarithmic-phase KJ cells was extracted and reverse-transcribed to cDNA using the primer FliC3-C (Appendix A). The resultant cDNA was used as the template for PCR with the primers FliC1Q178-F/R and FliC2Q163-F/R (Appendix A). SmeX, intrinsically unexpressed in KJ cells [36], was used as a negative control for DNA contamination during RNA preparation. PCR amplicons were separated by agarose gel electrophoresis and visualized by ethidium bromide staining.

### 4.3. Construction of Promoter-xylE Transcriptional Fusion Constructs

The 504-, 431-, and 388-bp DNA fragments upstream of *fliC1*, *fliC2*, and *fliC3* genes (the gray bars in the Figure 2A) were obtained by PCR using primer sets FliC1N-F/R, FliC1C-F/R, and FliC2C-F/R (Appendix A), respectively. Three PCR amplicons were cloned into pRKxylE [36], a *xylE* reporter plasmid, to generate pFliC1_xylE_, pFliC2_xylE_, and pFliC3_xylE._ Plasmids were mobilized into *S. maltophilia* KJ for C23O activity assay.

### 4.4. Catechol 2,3-Dioxygenase (C23O) Activity Determination

C23O activity was measured as previously described [43]. One unit of enzyme activity (U) was defined as the amount of enzyme that converts 1 nmole catechol per min. The specific activity of the enzyme (U/OD_450nm_) was defined in terms of units per 3.6 × 10^8^ cells (assuming that OD_450nm_ of 1 corresponds to 3.6 × 10^8^ cells/mL).

### 4.5. Construction of in-Frame Deletion Mutants

Double cross-over homologous recombination was used to create in-frame deletion mutants. Two PCR amplicons of 400–500 bp upstream and downstream of the intended deletion region were obtained by PCR and subsequently cloned into pEX18Tc. The primers used were FliC1N-F/FliC1N-R and FliC1C-F/FliC1C-R for pΔFliC1; FliC1C-F/FliC1C-R and FliC2C-F/FliC2C-R for pΔFliC2; as well as FliC2C-F/FliC2C-R and FliC3C-F/FliC3C-R for pΔFliC3 (Appendix A). These pEX18-derived plasmids, pΔFliC1, pΔFliC2, and pΔFliC3 (Appendix A), were used for mutant construction. Plasmid conjugation, transconjugants selection, and mutant confirmation were carried out as previously described [44]. The double mutant and triple mutant were sequentially constructed from the single mutant through the same procedure.

### 4.6. Swimming Motility Assay

Five microliters of logarithmically grown bacterial culture was applied onto the swimming agar (1% tryptone, 0.5% NaCl, and 0.15% agar) without or with additives. The additives included 12 μg/mL menadione (MD), 20 μg/mL 2,2′-dipyridyl (DIP), or 10% Ficoll. The plates were incubated at 37 °C for 48 h. The diameters (millimeters) of swimming zones were recorded.

### 4.7. Swarming Motility Assay

Five microliters of logarithmically grown bacterial culture was applied onto the swarming agar (1% tryptone, 0.5% NaCl, and 0.5% agar). The plates were sealed to maintain the humidity and incubated at 30 °C for 5 days. The diameters (millimeters) of swarming zones were recorded. Experiments were done in triplicate.

### 4.8. Flagella staining and TEM observation

Overnight-cultured bacterial cells were inoculated into fresh LB broth at an initial OD_450nm_ of 0.15 and then cultured for 5 h. Bacterial cells were collected, washed twice with PBS (pH 7.4), and then resuspended in PBS. Bacteria were negatively stained with 1% phosphotungstic acid (pH 7.4) on Formvar-coated copper grids and observed by transmission electron microscopy (TEM) (Hitachi H-7650, Hitachi, Tokyo, Japan) as previously described [13].

### 4.9. Adhesion on Abiotic Surfaces

Overnight-cultured bacteria were adjusted to a concentration of 5 × 10^8^ CFU/mL. Bacterial aliquot of 1 mL was inoculated into a 24-well flat-bottom microplate and quiescently incubated at 37 °C for 2 h. The non-adherent bacteria were washed with PBS, and the adherent cells were quantified by CFU counting.

### 4.10. Biofilm Formation on Abiotic Surfaces

Overnight-cultured bacteria were adjusted to a concentration of 10^5^ CFU/mL. Bacterial aliquot of 200 μL was inoculated into a 96-well microplate and quiescently incubated at 37 °C for 48 h. After removing the planktonic cells, the biofilm formed on the walls of the microplate wells were strained with 0.1% crystal violet for 20 min. Then, the crystal violet-stained wells of the microplates were washed with sterile water to remove excess dyes. Dyes of biofilms were dissolved by 30% acetic acid and quantified by measuring spectrophotometrically at 550 nm. Uninoculated medium controls were included.

### 4.11. Adhesion and Biofilm Formation on Biotic Surfaces

The ability of bacterial strains to adhere to and form biofilm on 293T cells was investigated using a static co-culture system as previously described [30] with some modification. The 293T cell was cultured in DMEM medium with 10% FBS, 50 units/mL penicillin, and 50 μg/mL streptomycin. They were cultured in 24-well polystyrene plates seeded with 5 × 10^5^ cells per well, grown to confluence in DMEM medium at 37 °C and 5% CO_2_, and infected with the analyzed *S. maltophilia* strain to obtain a multiplicity of infection (MOI) of 1000. After a 2- (for adhesion assay) and 24-h (for biofilm assay) incubation, respectively, cells were washed three times with PBS to removed non-adherent bacteria. Cells were removed by repeatedly pipetting, and the bacteria were plated on MH agar for CFU determination.

### 4.12. Bioinformatics Analysis

Protein sequence alignments were performed using the COBALT multiple alignment tool from NCBI “COBALT multiple alignment. Available online: https://www.ncbi.nlm.nih.gov/tools/cobalt/re_cobalt.cgi (accessed on 27 January 2022). The phylogeny tree was generated using the neighbor-joining method. We performed 1000 bootstrap inferences.

## Figures and Tables

**Figure 1 ijms-23-03863-f001:**
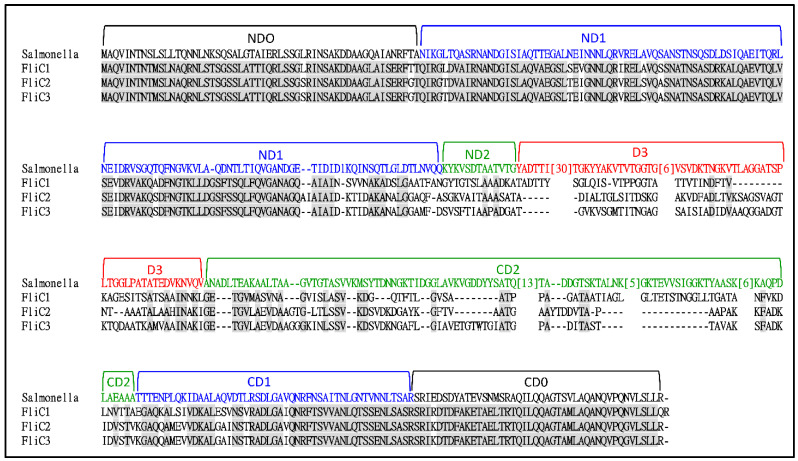
Multiple sequence alignment of *S. maltophilia* FliC1, FliC2, FliC3 proteins and *Salmonella* typhimurium FliC (Accession No. ALE 62494). Amino acid residues conserved across the FliC1, FliC2, and FliC3 proteins are marked in gray. The designation of domains is referenced from Namba’s group [32,33]. The numbers in the square brackets indicate the numbers of amino acid residues that were omitted.

**Figure 2 ijms-23-03863-f002:**
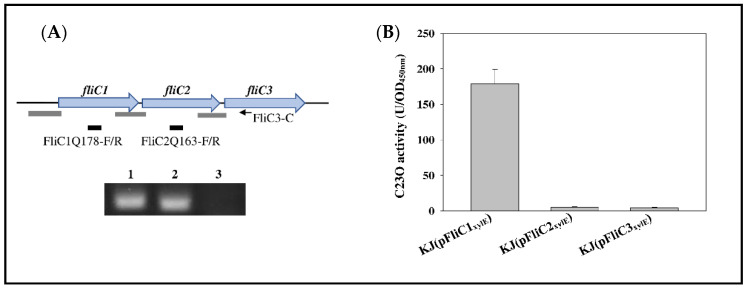
Characterization of *fliC1C2C3* operon of *S. maltophilia*: (**A**) The genetic organization of *fliC1C2C3* operon and agarose gel electrophoresis of PCR amplicon. The black arrow indicates the location of primer FliC3-C used for RT-PCR. Black bars represent the locations of PCR amplicons primered by FliC1Q178-F/R and FliC2Q163-F/R, respectively. Gray bars represent the locations of DNA segments for the constructions of pFliC1_xylE_, pFliC2_xylE_, and pFliC3_xylE_, which were used for the promoter activity assay in (**B**). DNA-free RNA was purified from logarithmically grown KJ cells, and cDNAs were obtained by RT-PCR using the primer FliC3-C. The cDNA was used as a template for PCR with the primers as indicated. Lane 1, primers FliC1Q178-F and FliC1Q178-R; Lane 2, primers FliC2Q163-F and FliC2Q163-R; Lane 3, primers SmeXQ-F and SmeXQ-R. SmeX, intrinsically unexpressed in KJ cells, was used as a negative control for DNA contamination during RNA preparation. (**B**) Promoter activity assay of *fliC1C2C3* operon. Overnight cultures of KJ(pFliC1_xylE_), KJ(pFliC2_xylE_), and KJ(pFliC3_xylE_) were inoculated into fresh LB with an initial OD_450nm_ of 0.15. Cells were grown aerobically for 5 h before measuring the C23O activity. Data are the means from three independent experiments. Error bars represent the standard deviations for three triplicate samples. Significance calculated by Student’s *t*-test.

**Figure 3 ijms-23-03863-f003:**
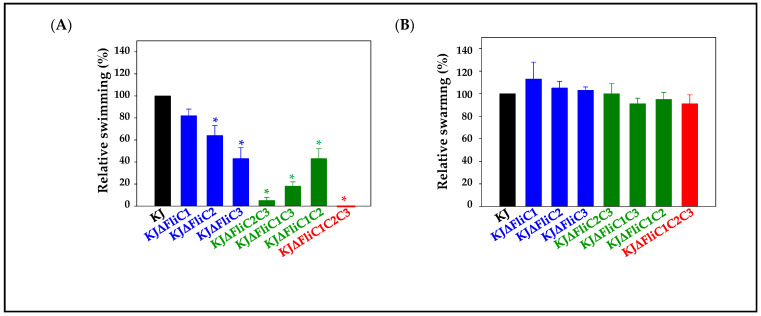
The swimming, swarming, flagellum morphology, adhesion, and biofilm formation of wild-type KJ and its derived *fliC*-associated mutants. The color coding for black, blue, green, and red represent wild-type, single-deletion mutants, double-deletion mutants, and triple-deletion mutant. Data are the means from three independent experiments. Error bars indicate the standard deviations for three triplicate samples. *, *p* < 0.05, significance calculated by Student’s *t*-test. (**A**) Swimming motility. Five microliters of overnight-cultured bacterial cell suspension was inoculated into swimming agar and then incubated at 37 °C for 48 h. The swimming zones were recorded. (**B**) Swarming motility. Five microliters of overnight-cultured bacterial cell suspension was inoculated into swarming agar and then incubated at 30 °C for 5 days. The swarming zones were recorded. (**C**) Flagella morphology. Overnight-cultured bacterial cells were inoculated into fresh LB broth and then grown for 5 h. The flagella were negatively stained with 1% phosphotungstic acid (pH 7.4) and observed by TEM. The average flagellum numbers per cell (at least 10 cells in each condition) and flagellum length (at least 20 flagella in each condition) were calculated. (**D**) Adhesion ability on abiotic surfaces. Tested *S. maltophilia* strain was adjusted to the concentration of 5 × 10^8^ CFU/mL. Bacterial aliquot of 1 mL was inoculated into 24-well flat-bottom microplate and incubated at 37 °C for 2 h. The non-adherent bacteria were removed by PBS wash, and the adherent bacteria were quantified by CFU determination. (**E**) Biofilm formation on abiotic surfaces. Biofilm formation capacity of each strain was tested by the ability of the cells to adhere to the 96-well microplates, followed by crystal violet staining. (**F**,**G**) Adhesion ability and biofilm formation on biotic surfaces. The 293T cells were infected with analyzed *S. maltophilia* strain to obtain a multiplicity of infection (MOI) of 1000. After a 2-h (for adhesion ability) and 24-h (for biofilm formation) incubation, respectively, the non-adherent bacteria were removed by PBS wash, and the adherent bacteria were quantified by CFU determination.

**Figure 4 ijms-23-03863-f004:**
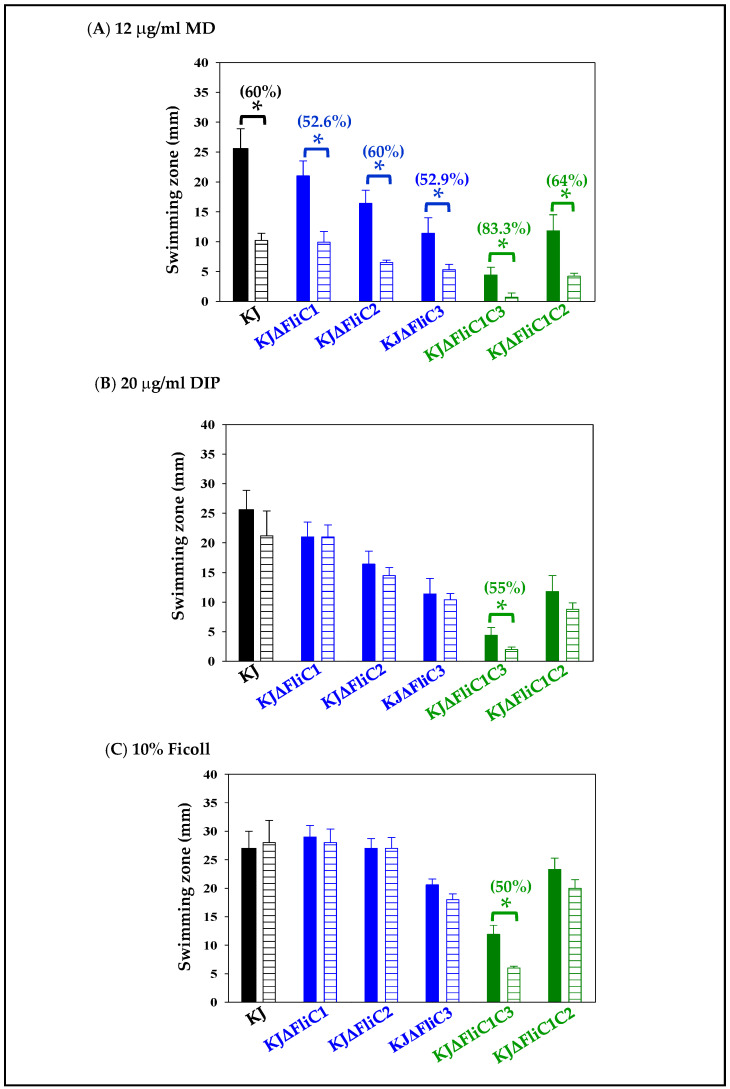
Roles of the three flagellin proteins in swimming motility under different stressed environments. Five microliters of overnight-cultured bacterial cell suspension was inoculated into swimming agar without and with different additives, and then incubated at 37 °C for 48 h. Swimming zones were recorded. Solid bars indicate the condition with no additive, and striped bars indicate the conditions with stressor additives. The numbers labelled above the bars indicate the swimming compromise level under the stress conditions. Swimming-compromise level was calculated using the equation: [100–(swimming zone in additive-containing agar/swimming zone in additive-null agar) × 100]. The additives included 12 μg/mL menadione (MD) (**A**), 20 μg/mL 2,2′-dipyridyl (DIP) (**B**), and 10% Ficoll (**C**). Data are the means from three independent experiments. Error bars indicate the standard deviations for three triplicate samples. *, *p* < 0.05, significance calculated by Student’s *t*-test.

**Figure 5 ijms-23-03863-f005:**
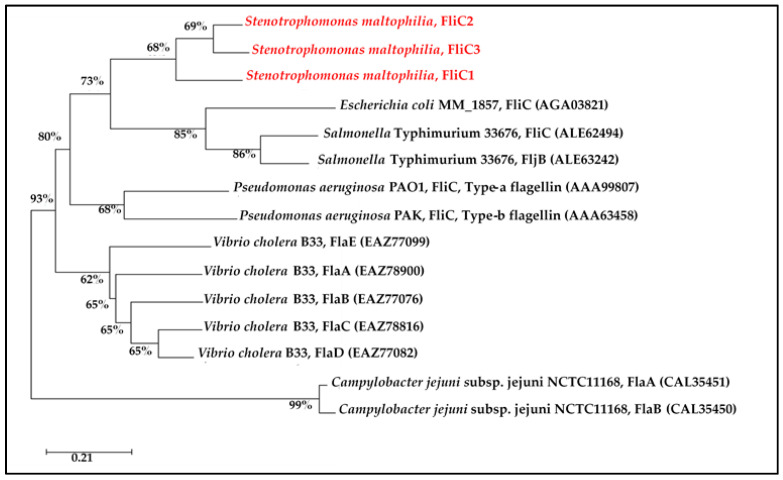
Phylogenetic analysis of the flagellins of *S. maltophilia* KJ and other bacterial species. The phylogeny was created with the flagellin amino acid sequences using the neighbor-joining method. Numbers at the branch nodes indicate the bootstrap values as a percentage of 1000 replications. The accession numbers of the proteins are listed in the brackets.

## Data Availability

Not applicable.

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
