# Peer review of "Molecular Characterization of Three Tandemly Located Flagellin Genes of Stenotrophomonas maltophilia"

_ijms, 2022, doi:10.3390/ijms23073863_

Round 1
Reviewer 1 Report
In this paper, titled “Molecular characterization of three tandemly located flagellin genes of Stenotrophomonas maltophilia”, Wu et al, aim to describe the role of the three flagellin genes in the flagella motility and synthesis in S. maltophilia. This work is interesting but in my eyes, the mutants are not exploited, as they should. More work about phenotypic characterization should be included. Important papers describing S. maltophilia flagella function are not cited. For these reasons, the paper need to be improved before acceptance.
The authors discussed in the introduction about flagellar adhesion and the impact of this adhesion on bacterial pathogenicity. The authors are only referring to S. maltophilia and then should add more papers regarding flagella adhesion in other bacterial species to broaden the paper's interest. Likewise, the authors described interesting phenotypes of the different mutants regarding their flagellation but focus only on biofilm formation to link the flagella to adhesion. Flagella can play a role in biofilm initiation but as well as in biofilm maturation. Therefore, proper adhesion assays on abiotic or biotic surfaces need to be added for the readership.
The authors are not testing the swarming motility of the different mutants, which is dependent on the number of flagella. It will be an asset to the paper to measure this phenotype.
Bacterial flagellin of E. coli or Salmonella spp. have long and very variable domains D2-D3 so which strains have been used for the phylogenetic analysis for these bacteria? What type of results the authors would have obtained with different H serotypes for this tree?
In figure 1, the different flagellin domains should be highlighted.
The papers from the Di Bonaventura lab are not cited in this paper. This lab has published many works on Stenotrophomonas maltophilia and then they should be discussed.
Did the authors are aware of the post-translational modification of Stenotrophomonas maltophilia flagellins?
Line 320: Need to rephrase.
Author Response
Reviewer 1
In this paper, titled “Molecular characterization of three tandemly located flagellin genes of Stenotrophomonas maltophilia”, Wu et al, aim to describe the role of the three flagellin genes in the flagella motility and synthesis in S. maltophilia. This work is interesting but, in my eyes, the mutants are not exploited, as they should. More work about phenotypic characterization should be included. Important papers describing S. maltophilia flagella function are not cited. For these reasons, the paper need to be improved before acceptance.
The authors discussed in the introduction about flagellar adhesion and the impact of this adhesion on bacterial pathogenicity. The authors are only referring to S. maltophilia and then should add more papers regarding flagella adhesion in other bacterial species to broaden the paper's interest. Likewise, the authors described interesting phenotypes of the different mutants regarding their flagellation but focus only on biofilm formation to link the flagella to adhesion. Flagella can play a role in biofilm initiation but as well as in biofilm maturation. Therefore, proper adhesion assays on abiotic or biotic surfaces need to be added for the readership.
Reply: The references regarding flagella adhesion in other bacterial species have been added. The adhesion assays on abiotic or biotic surfaces have been included. Please see Lines 41, 206-214, 319-326, 453-461, 528-532, 542-552, references 5-6, and Figure 3D-3G in the revised manuscript.
The authors are not testing the swarming motility of the different mutants, which is dependent on the number of flagella. It will be an asset to the paper to measure this phenotype.
Reply: The result of swarming motility was included. Please see Lines 195-196, 314-316, 516-520, and Figure 3B in the revised manuscript.
Bacterial flagellin of E. coli or Salmonella spp. have long and very variable domains D2-D3 so which strains have been used for the phylogenetic analysis for these bacteria? What type of results the authors would have obtained with different H serotypes for this tree?
Reply: The bacteria strains and the accession numbers of flagellin proteins used in the phylogenetic analysis have been included in phylogenetic analysis. The outcome of Salmonella with different H serotypes to phylogenetic tree has been described. Please see Lines 415-419, 424-425, and Figure 5 in the revised manuscript.
In figure 1, the different flagellin domains should be highlighted.
Reply: The domains of FliC protein have been highlighted. Please see Lines 123-127, 131-134, and Figure 1 in the revised manuscript.
The papers from the Di Bonaventura lab are not cited in this paper. This lab has published many works on Stenotrophomonas maltophilia and then they should be discussed.
Reply: The papers from Di Bonaventura’s group have been cited and discussed. Please see Lines 103, 453-455 and references 30-31 in the revised manuscript.
Did the authors are aware of the post-translational modification of Stenotrophomonas maltophilia flagellins?
Reply: The issue concerning the post-translational modification of S. maltophilia has not been reported so far. The flagellin post-translational modification is common in Gram-negative bacteria; thus, we did not rule out the possibility.
Line 320: Need to rephrase.
Reply: The sentence has been rephrased. Please see Lines 463-465 in the revised manuscript.
Reviewer 2 Report
The manuscript by Wu et al. investigates the role of three flagellin genes in swimming motility and biofilm formation of Stenotrophomonas maltophilia. Mutants with single and multiple flagellin gene deletions had varied effects on swimming motility, biofilm formation, and flagellar filament morphology. These findings are interesting, but additional experiments are necessary to substantiate the conclusion drawn in this work.
Major comments:
- The cell culture supernatant in the different flagellin mutants should be checked for sheared flagellar filaments and free flagellin to determine whether the mutations have an effect on flagellar filament stability and integrity.
- 3B: Why are no standard deviations listed for flagellar length of four of the strains? The statistical significance of the data in the Table is missing. Information about the number of cells analyzed is missing. The numbers above the length bars in the EM images are ineligible.
- 4: The cross-reactivity western blot is not a meaningful result. It is to be expected that a polyclonal antiserum against one flagellin protein from one species would cross-react with other flagellin proteins from the same species due to the highly conserved N- and C-terminal regions. This figure should be omitted.
- How was it determined that the three different additives actually mimic different environmental conditions? The concentration of glycerol does not create a high enough viscosity to evaluate its effect on flagellar motility. Typically, other agents, such as Ficoll are being used for these types of measurements. A dose-response curve with the wild type would have been useful to determine the actual testing concentrations. Since there was either no effect or a similar effect on the deletion mutants versus wild type, there is not much of a conclusion that can be drawn from this experiment.
- 6: It is unclear why this figure is depicted in the discussion. It should be moved to Supplemental Material.
- Line 286-290: The conclusion that the weaker band intensity is due to weak cross-reactivity is incorrect. It is also possible that the weaker band is due to a lower amount of overall flagellin protein. It is known that negative feed-back regulation exists if non/poorly-functional combinations of flagellin proteins are being expressed. It is also unclear how this feature would enable evasion of the immune system.
- Line 293-296: The statement that flagellin proteins are redundant in biofilm formation is incorrect. The data show that the expression of flagellins and the presence of at least one shortened flagellum is sufficient to promote biofilm formation.
- Overall, the study appears incomplete since is presents flaws in the interpretation of data, the lack of appropriate control experiments, and unjustified conclusions. Exploration of the expression levels of the three flagellin proteins would have been informative and would shed light on the role of individual flagellins in the wild type and deletion mutant strains.
Minor comments:
- Line 16: Should read ‘The flagellum, which.’
- Line 21: Should read ‘and a triple mutant.’
- Line 26: Should read ‘only the triple mutant.’
- Line 28: Should read ‘The three flagellins.’
- Line 28: Specify the meaning of ‘shared certain epitopes.’
- Line 29: Specify ‘antigenicity seemed to be weak.’
- Line 227: Should read ‘overnight culture.’
- Line 233/234: Should read ‘of the presence of three different.’
- Line 235: Should read ‘to form a filament.’
- Line 280: Not the N- and C-termini are conserved but the N- and C-terminal regions or domains.
- Line 336: Should read ‘Flagellin appears to be’ and ‘for the development of an inhibitor.’
- Line 338-341: Sentence should be rephrased for logical purposes.
- Line 377: Define ‘MD’ and ‘DIP.’
- Line 386: The unit after ‘200’ is incomplete.
- 2: Fig. 2A and Fig. 2B have not been labeled accordingly.
- 2: The bands in the RT-PCR image are not very well contrasted and hard to see.
- 5: MD and DIP should be defined in the figure legend.
- Table S2: The delta symbol should not be italicized.
Author Response
Reviewer 2
The manuscript by Wu et al. investigates the role of three flagellin genes in swimming motility and biofilm formation of Stenotrophomonas maltophilia. Mutants with single and multiple flagellin gene deletions had varied effects on swimming motility, biofilm formation, and flagellar filament morphology. These findings are interesting, but additional experiments are necessary to substantiate the conclusion drawn in this work.
Major comments:
- The cell culture supernatant in the different flagellin mutants should be checked for sheared flagellar filaments and free flagellin to determine whether the mutations have an effect on flagellar filament stability and integrity.
Reply: Your suggestion has been considered and sheared flagellar filaments in the culture supernatant of mutants has been rechecked by TEM observation. Please see Lines 202-204 in the revised manuscript.
- 3B: Why are no standard deviations listed for flagellar length of four of the strains? The statistical significance of the data in the Table is missing. Information about the number of cells analyzed is missing. The numbers above the length bars in the EM images are ineligible.
Reply: According to your suggestions, we have made the corrections. Please see Figure 3C and Lines 318-319 in the revised manuscript.
- The cross-reactivity western blot is not a meaningful result. It is to be expected that a polyclonal antiserum against one flagellin protein from one species would cross-react with other flagellin proteins from the same species due to the highly conserved N- and C-terminal regions. This figure should be omitted.
Reply: The figure 4 in the previous manuscript has been deleted.
- How was it determined that the three different additives actually mimic different environmental conditions? The concentration of glycerol does not create a high enough viscosity to evaluate its effect on flagellar motility. Typically, other agents, such as Ficoll are being used for these types of measurements. A dose-response curve with the wild type would have been useful to determine the actual testing concentrations. Since there was either no effect or a similar effect on the deletion mutants versus wild type, there is not much of a conclusion that can be drawn from this experiment.
Reply: Thank you for your suggestions. Experiment concerning the impact of Ficoll on swimming has been added accordingly. Please see Figure 4C, Figure S2, and Lines 338-342 in the revised manuscript.
- 6: It is unclear why this figure is depicted in the discussion. It should be moved to Supplemental Material.
Reply: Integrating the suggestions from reviewer 1 and reviewer 2, we had moved Figure 5 to the Results section. Please see Figure 5 and Lines 407-419 in the revised manuscript.
- Line 286-290: The conclusion that the weaker band intensity is due to weak cross-reactivity is incorrect. It is also possible that the weaker band is due to a lower amount of overall flagellin protein. It is known that negative feed-back regulation exists if non/poorly-functional combinations of flagellin proteins are being expressed. It is also unclear how this feature would enable evasion of the immune system.
Reply: According to your suggestion in the major comment 3, the figure 4 in the previous manuscript has been deleted accordingly. Thus, these sentences have been deleted.
- Line 293-296: The statement that flagellin proteins are redundant in biofilm formation is incorrect. The data show that the expression of flagellins and the presence of at least one shortened flagellum is sufficient to promote biofilm formation.
Reply: Thank for your comments. Based on the suggestions from reviewer 1, the biofilm experiment has been extended and repeated. Please see Figure 3E and 3G, and Lines 458-461 in the revised manuscript.
- Overall, the study appears incomplete since is presents flaws in the interpretation of data, the lack of appropriate control experiments, and unjustified conclusions. Exploration of the expression levels of the three flagellin proteins would have been informative and would shed light on the role of individual flagellins in the wild type and deletion mutant strains.
Reply: Thank for your comments. Based on three reviewers’ suggestions, some experiments have been modified or extended and new analyses have been added in the revised manuscript.
Minor comments:
- Line 16: Should read ‘The flagellum, which.’
Reply: The sentence has been corrected. Please see Line 16 in the revised manuscript.
- Line 21: Should read ‘and a triple mutant.’
Reply: The sentence has been corrected. Please see Line 21 in the revised manuscript.
- Line 26: Should read ‘only the triple mutant.’
Reply: According to your suggestion in the major comment 3, the results of figure 4 in the previous manuscript has been deleted; thus, the sentence has been deleted.
- Line 28: Should read ‘The three flagellins.’
Reply: According to your suggestion in the major comment 3, the results of figure 4 in the previous manuscript has been deleted; thus, the sentence has been deleted.
- Line 28: Specify the meaning of ‘shared certain epitopes.’
Reply: According to your suggestion in the major comment 3, the results of figure 4 in the previous manuscript has been deleted; thus, the sentence has been deleted.
- Line 29: Specify ‘antigenicity seemed to be weak.’
Reply: According to your suggestion in the major comment 3, the results of figure 4 in the previous manuscript has been deleted; thus, the sentence has been deleted.
- Line 227: Should read ‘overnight culture.’
Reply: According to your suggestion in the major comment 3, the results of figure 4 in the previous manuscript has been deleted; thus, the sentence has been deleted.
- Line 233/234: Should read ‘of the presence of three different.’
Reply: The sentence has been corrected. Please see Lines 329-330 in the revised manuscript.
- Line 235: Should read ‘to form a filament.’
Reply: The sentence has been corrected. Please see Line 331 in the revised manuscript.
- Line 280: Not the N- and C-termini are conserved but the N- and C-terminal regions or domains.
Reply: According to your suggestion in the major comment 3, the results of figure 4 in the previous manuscript has been deleted; thus, the sentence has been deleted.
- Line 336: Should read ‘Flagellin appears to be’ and ‘for the development of an inhibitor.’
Reply: The sentence has been corrected. Please see Line 472 in the revised manuscript.
- Line 338-341: Sentence should be rephrased for logical purposes.
Reply: The sentence has been rephrased. Please see Lines 475-477 in the revised manuscript.
- Line 377: Define ‘MD’ and ‘DIP.’
Reply: MD and DIP have been defined. Please see Line 513 in the revised manuscript.
- Line 386: The unit after ‘200’ is incomplete.
Reply: The typo has been corrected. Please see Line 535 in the revised manuscript.
- : Fig. 2A and Fig. 2B have not been labeled accordingly.
Reply: The Fig. 2A and Fig. 2B have been labeled accordingly. Please see Figure 2 in the revised manuscript.
- : The bands in the RT-PCR image are not very well contrasted and hard to see.
Reply: We have re-performed this experiment and a better image was presented in Fig. 2A.
- : MD and DIP should be defined in the figure legend.
Reply: MD and DIP have been defined in the figure legend of figure 2. Please see Line 404 in the revised manuscript.
- Table S2: The delta symbol should not be italicized.
Reply: The mistakes have been corrected. Please see Table S2 in the revised manuscript.
Reviewer 3 Report
The manuscript focuses on the characterization of fliC1, fliC2, and fliC3 in Stenotrophomonas maltophilia. Expression of the fliC operon is investigated using promoter constructs and clean deletions in fliC1, fliC2, and fliC3 (single, double and triple mutants). Mutants were texted and compared to WT for biofilm formation, swimming motility, fliC protein expression, flagella per cell and flagella length. Overall the manuscript is well written and straight forward.
Specific comments:
Figure 2. Individual images lack lack notation of A, B, and C. Also state what error bars represent.
Figure 3: how many cells were counted to determine flagella per cell? How many flagella were measured for length in each condition? The color coding for black/blue/green is not explained? Numbers above scale bars are not readable when zoomed in.
Phylogenetic analysis (Figure 6) presented in the discussion should be moved into results. The discussion should be reserved for discussing the existing results and not introducing new information. Figure 6 is also low resolution and needs to be replaced.
The discussion in the current draft is more or less a reiteration of the results and there is limited discussion of how the data obtained actually fits into the bigger picture and relevant literature from related organisms. The discussion needs to be revamped with an improved discussion comparing results obtained in this paper with others.
Bioinformatic analysis is missing in the methods and needs to be added. The exact strains from which sequences were obtained and their corresponding gene accession numbers should also be included for the generation of Figure 6.
Line 153: Change "No polar effect " to "A lack of polar effects"
Line 215: Student’s test should be "Student’s t test"
Line 267: varies should be vary
Line 377: Define MD and DIP
Lines 380/381: How were the cultures collected and prepared for microscopy?
Line 386: "aliquot of 200 l was inoculated" something is wrong here. please edit.
Line 393: Describe how the cells were collected for protein isolation
Line 397: Where were the anti-FliC3 and anti-RpoA antibodies purchased or where were they obtained from?
Author Response
Reviewer 3
Comments and Suggestions for Authors
The manuscript focuses on the characterization of fliC1, fliC2, and fliC3 in Stenotrophomonas maltophilia. Expression of the fliC operon is investigated using promoter constructs and clean deletions in fliC1, fliC2, and fliC3 (single, double and triple mutants). Mutants were texted and compared to WT for biofilm formation, swimming motility, fliC protein expression, flagella per cell and flagella length. Overall the manuscript is well written and straight forward.
Specific comments:
Figure 2. Individual images lack notation of A, B, and C. Also state what error bars represent.
Reply: The notion of Figure 2 and the statement about error bars have been added. Please see Figure 2 and Lines 180-182 in the revised manuscript.
Figure 3: how many cells were counted to determine flagella per cell? How many flagella were measured for length in each condition? The color coding for black/blue/green is not explained? Numbers above scale bars are not readable when zoomed in.
Reply: Numbers of the countered cells and flagella have been indicated in the figure legend of Figure 3. The color meanings in the Figure 3 have been explained. The quality of figure 3C has been improved. Please see Figure 3 and Lines 310-311 & 318-319 in the revised manuscript.
Phylogenetic analysis (Figure 6) presented in the discussion should be moved into results. The discussion should be reserved for discussing the existing results and not introducing new information. Figure 6 is also low resolution and needs to be replaced.
Reply: Figure 5 (Figure 6 in the previous manuscript) has been presented in the Results section and its resolution has been improved. Please see Figure 5 in the revised manuscript.
The discussion in the current draft is more or less a reiteration of the results and there is limited discussion of how the data obtained actually fits into the bigger picture and relevant literature from related organisms. The discussion needs to be revamped with an improved discussion comparing results obtained in this paper with others.
Reply: Thank for your comments. Based on three reviewers’ suggestions, some experiments have been modified or extended and new analyses have been added in the revised manuscript. In addition, the discussion is also revamped.
Bioinformatic analysis is missing in the methods and needs to be added. The exact strains from which sequences were obtained and their corresponding gene accession numbers should also be included for the generation of Figure 6.
Reply: The methods for bioinformatic analysis have been included in the Materials and Methods section. The bacteria strains and the accession numbers of flagellin proteins used in the phylogenetic analysis (Figure 5) have been included. Please see Lines 424-425, 554-558 and Figure 5 in the revised manuscript.
Line 153: Change "No polar effect " to "A lack of polar effects"
Reply: The words have been edited accordingly. Please see Lines 188-189 in the revised manuscript.
Line 215: Student’s test should be "Student’s t test"
Reply: The typo has been edited accordingly. Please see Line 312 in the revised manuscript.
Line 267: varies should be vary
Reply: The word has been edited accordingly. Please see Line 408 in the revised manuscript.
Line 377: Define MD and DIP
Reply: The MD and DIP have been defined accordingly. Please see Line 513 in the revised manuscript.
Lines 380/381: How were the cultures collected and prepared for microscopy?
Reply: The method for culture collection has been included. Please see Lines 522-523 in the revised manuscript.
Line 386: "aliquot of 200 l was inoculated" something is wrong here. please edit.
Reply: Thanks for your kind reminder. The error has been edited. Please see Line 535 in the revised manuscript.
Line 393: Describe how the cells were collected for protein isolation
Reply: According to the suggestions from reviewer 2, the results of figure 4 in the previous manuscript has been deleted; thus, the sentence has been deleted.
Line 397: Where were the anti-FliC3 and anti-RpoA antibodies purchased or where were they obtained from?
Reply: According to the suggestions from reviewer 2, the results of figure 4 in the previous manuscript has been deleted; thus, the sentence has been deleted.
Round 2
Reviewer 1 Report
While the authors had answered most of my questions, an important aspect remains to be improved regarding the discussion of the results. What are the hypotheses that the authors have in mind about the different phenotypes of each mutant? For example, why does FliC3 seem to play an important role in biofilm formation? Why the flagellin expression is not affecting the swarming (The swarming results are not discussed)? Did the authors have measured the flagella diameter for each mutant in figure 3?
Author Response
While the authors had answered most of my questions, an important aspect remains to be improved regarding the discussion of the results. What are the hypotheses that the authors have in mind about the different phenotypes of each mutant? For example, why does FliC3 seem to play an important role in biofilm formation? Why the flagellin expression is not affecting the swarming (The swarming results are not discussed)? Did the authors have measured the flagella diameter for each mutant in figure 3?
Reply:
- The explanations for the role of FliC3 in biofilm formation have been included. Please see Lines 461-467 in the revised manuscript.
- Discussion of swarming results have been included. Please see Lines 468-480 in the revised manuscript.
- The flagella diameters for each mutant in figure 3 were not detected in this study.
Reviewer 2 Report
Major comments:
- The cell culture supernatant in the different flagellin mutants should be checked for secreted flagellin monomers to determine whether the mutations have an effect on flagellar filament stability and integrity. A western blot of cell culture supernatants should be performed to analyze whether flagellin monomers are being secreted but not assembled.
- The study improved due to the omission of incomplete and inconclusive data but still fails to contribute new insights into the function of bacterial flagella.
Minor comments:
- Line 25/26: Should read ‘The flagellin triple mutant’ or “The flagellin null mutant’ and ‘and was compromised.’
- Line 30: I don’t think that the term ‘superior’ should be used in this context. It would be better to say ‘flagella adapted to provide better ability…’
Reviewer 3 Report
I am satisfied that all of my concerns have been addressed
Author Response
Thank you very much.
Round 3
Reviewer 2 Report
Line 204: Free flagellin proteins cannot be detected by electron microscopy. This statement need to be removed.
Author Response
Line 204: Free flagellin proteins cannot be detected by electron microscopy. This statement need to be removed.
Reply: Thanks for your comments. The sentence has been removed.